

# Design and analysis of teaching early warning system based on multimodal data in an intelligent learning environment

Xinxin Kang[1,2] and Yong Nie[1]

[1] School of Education, Shaanxi Normal University, Xi'an, Shaanxi, China
[2] Informatization Office, Xi'an Conservatory of Music, Xi'an, Shaanxi, China

## ABSTRACT

In online teaching environments, the lack of direct emotional interaction between teachers and students poses challenges for teachers to consciously and effectively manage their emotional expressions. The design and implementation of an early warning system for teaching provide a novel approach to intelligent evaluation and improvement of online education. This study focuses on segmenting different emotional segments and recognizing emotions in instructional videos. An efficient long-video emotional transition point search algorithm is proposed for segmenting video emotional segments. Leveraging the fact that teachers tend to maintain a neutral emotional state for significant portions of their teaching, a neutral emotional segment filtering algorithm based on facial features has been designed. A multimodal emotional recognition model is proposed for emotional recognition in instructional videos. It begins with preprocessing the raw speech and facial image features, employing a semi-supervised iterative feature normalization algorithm to eliminate individual teacher differences while preserving inherent differences between different emotions. A deep learning-based multimodal emotional recognition model for teacher instructional videos is introduced, incorporating an attention mechanism to automatically assign weights for feature-level modal fusion, providing users with accurate emotional classification. Finally, a teaching early warning system is implemented based on these algorithms.

## INTRODUCTION

In recent years, particularly following the global spread of the COVID-19 pandemic, many schools have adopted online classrooms and explored blended learning models combining online and offline teaching methods (*Nuypukiaw & Chompurach, 2023*; *Suwathanpornkul et al., 2023*). However, the lack of face-to-face emotional interaction between teachers and students in online settings poses a challenge (*Pu & Xu, 2023*). While some studies have focused on tracking learners' real-time emotional states during online classes, teachers also play a crucial role in teaching (*Turner & Hodis, 2023*). Teachers often utilize exaggerated expressions and tones of happiness, surprise, frustration, and so on to regulate the classroom atmosphere and emphasize key points. Simultaneously, teachers may also

Corresponding author
Yong Nie, sdkxx2022@163.com

experience negative emotions due to their own stress or anxiety (*Hui, 2021*). In live or recorded teaching, the lack of direct emotional exchange between teachers and students makes it difficult for teachers to consciously and effectively manage their emotional expressions (*Farrell, 2021*; *Gilmore, 2007*). Therefore, studying the emotional variations in teachers' online videos can pave a new way for evaluating the effectiveness of intelligent classrooms in online education.

Currently, there is a scarcity of research using artificial intelligence techniques to specifically measure teachers' teaching emotions. Consequently, we will draw inspiration from the methods used to recognize learners' emotions in online classrooms (*Koizumi et al., 2023*; *Zhou, Yan & Shang, 2016*). Since it is challenging to identify body movements in online teaching, most studies utilize learners' facial images or facial features as input. The research on emotional recognition in teachers' instructional videos can be informed by methods used for recognizing learners' facial expressions. However, teachers occupy a central position as the primary speaker during the instruction, making their audio an equally important indicator of emotional changes (*Janice, 2023*; *Kladder, 2021*).

Additionally, for long videos of teachers' instructional sessions, which often encompass multiple emotions, it is necessary first to apply emotion transition point recognition techniques to identify these points and segment the long videos into shorter clips containing a single emotion (*Reed et al., 2023*; *Gaggioli, 2019*; *Dong et al., 2024*). This allows for the application of the techniques above for emotion recognition. A neutral emotional segment is defined as those parts of an instructional video that are characterized by a rather neutral emotion displayed by the teacher, based, more specifically, on data derived from facial and voice measurements. These segments are used as currents to find out more detailed emotional changes.

This study proposes a method to filter out extended neutral emotional segments in videos based on facial features. Then, it introduces a distance-based voice emotion transition point algorithm to segment the remaining video clips. The accuracy of this work is verified through experiments on standard datasets and applications to instructional videos. Furthermore, an algorithm is introduced for iteratively normalizing the original features based on the subject's neutral video segments to address the issue of eliminating subject-specific differences. The correctness of this work is validated through experiments and analysis. Finally, a teaching early warning system is designed and implemented.

## VIDEO EMOTION TRANSITION POINT SEARCH ALGORITHM

### Neutral emotion segment filtering algorithm based on facial features

Assuming a teacher has several teaching videos, we extract the first m frames from each video as neutral facial data, denoted as:

$$X = \{s_i | i = 1, 2, \ldots, N\} \tag{1}$$

where $s_i$ represents the feature vector of the $i^{th}$ facial image, and N represents the total number of neutral facial image frames for that teacher. We employ a mixture

density to model the characteristic distribution of an individual's neutral face (*Ulukaya & Erdem, 2012*):

$$p(s) = \sum_{k=1}^{k} p(s|G_k)P(G_k). \tag{2}$$

Here, k represents the number of Gaussian distributions, $G_k$ denotes the mixture components comprising all Gaussian distributions, $P(G_k)$ is the mixture coefficient, and $p(s|G_k)$ assumes that the feature vectors in the dataset follow an independent distribution. We utilize the Akaike Information Criterion (AIC) to select the optimal number of components. Once the GMM fitting is complete, the mean vectors of the k Gaussian mixture components, $\mu_k$, will represent the overall neutral facial characteristics of the teacher. Assuming these neutral facial feature vectors follow an independent distribution, the likelihood function of the samples is expressed as:

$$p(X|\Phi) = \Pi_{n=1}^{N}\left(\sum_{k=1}^{k} P(G_k)N\left(s_n|\mu_k, \sum_{k}\right)\right). \tag{3}$$

In addition, the AIC was used to determine the number of Gaussian components for GMMs to be used when selecting the right number of clusters. Decisions were made by comparing component values ranging from 1 to 21 to get the lower AIC, which gives the best model, not too complex or too simple. The primary study adopted the steady-state method, and in the dual-threshold method, the window sizes of 0.1–0.5 s were tried systematically. Such a time value of 0.3 s was chosen because this time provided the maximum classification accuracy while at the same time ensuring reasonable computational costs. Such a systematic approach guarantees the reliability and reproduction of the presented methods and their characteristics.

By utilizing the Expectation-Maximization (EM) algorithm, we maximize the log-likelihood function above, denoted as the maximum likelihood score:

$$score = \ln p(X|\Phi) = \sum_{n=1}^{N} \ln\left(\sum_{k=1}^{k} P(G_k)N\left(s_n|\mu_k, \sum_{k}\right)\right). \tag{4}$$

We can obtain the maximum likelihood score by substituting the facial features extracted from the remaining video frames with the above formula. If the score exceeds a predefined threshold, it is considered that the frame is likely to be a neutral expression frame.

The iterative feature normalization algorithm used one-class SVMs to classify the segments into neutral emotions. One-class SVMs are effective in the current situation given that the number of data samples for each class was relatively small and the data features were relatively easy to classify. However, in the case of bigger and more detailed datasets, more advanced methods, including an autoencoder for anomaly detection, are a far more appealing proposition. Autoencoders are good at scalability because they can

utilize neural networks' deep learning architecture to learn detailed data distribution patterns and variation qualifiers. As for this study, the selected one-class SVM produced stable results with good accuracy and little computational load; autoencoders may improve the stability and flexibility for subsequent larger-scale studies.

The neutral expression model mentioned above is trained solely using the initial frames of each video, resulting in stringent criteria for determining "neutral frames" due to the limited input data. Therefore, in this study, we utilize this model primarily to filter out neutral emotion segments for longer than 5 s while discarding shorter identified neutral segments to be processed in the next stage for more accurate emotional boundary identification. Specifically, we first extract the first m frames of each teaching video of a teacher and fit these frames to create the teacher's neutral facial model. Then, we sequentially input the remaining video frames into the model to calculate their maximum likelihood values. If more than 10 consecutive maximum likelihood values exceed a threshold, these frames are extracted as candidate intervals. Finally, we apply K-nearest neighbors (KNN) clustering to the candidate intervals, and intervals with a cluster size exceeding 500 frames are identified as neutral emotion segments.

## Emotion transition point search algorithm based on speech features

The process of rapid emotional transition point identification based on speech characteristics is illustrated in Fig. 1.

During speech, extended pauses may result in numerous silent intervals mixed within the initial speech segment. False emotional transition points are prone to detection at the junction between active speech and silent intervals, potentially interfering with experimental results. Therefore, this study employs a dual-threshold endpoint detection method that utilizes short-term energy and short-term average zero-crossing rate of the speech signal for effective speech endpoint detection. A speech segment comprises multiple silent and speech intervals, which contain unvoiced and voiced sounds. Unvoiced sounds, belonging to consonants in speech, typically occur after silent sections and before voiced sections. Since the energy of voiced sounds is higher than that of unvoiced sounds, and the zero-crossing rate of unvoiced sounds is higher than that of silent sections, we can first filter out the voiced sounds using an energy threshold and then extract the unvoiced sounds using the zero-crossing rate. This process allows us to segment the speech and silent intervals in the audio and remove the silent intervals. An effective speech segment is obtained after segmenting and concatenating the speech intervals. The openSmile 2.3 tool is then used to extract speech features, with the configuration file "MFCCI20DAn" selected for MFCC feature extraction and "prosodyShsconf" for prosodic feature extraction. The MFCC features utilized in this study comprise 13 base MFCC values calculated from 26 Mel filters and 13 first-order and 13 second-order derivative coefficients derived from these base values, totaling 39 features. The prosodic features include fundamental frequency (F0) voicing probability and loudness contours.

For an experimental audio clip, loudness values are extracted from the prosodic features every five units to plot a loudness contour. Subsequently, using the sound energy to define thresholds, the peaks and troughs of the entire loudness contour are roughly distinguished.

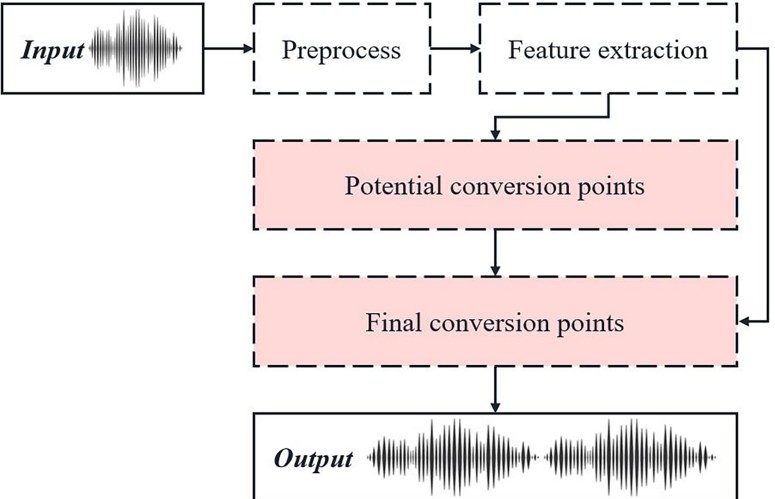

**Figure 1 Schematic diagram of fast emotion transition point recognition process based on speech features.**

Since a single threshold can easily contour that is continuously above or below the threshold, two thresholds are employed to divide the loudness contour into three segments. We define a central non-constant value u, representing the average intensity of the audio segment's energy, and σ, representing the variance of the audio segment's energy. From these, two boundary thresholds are generated:

$$Low_T h = \mu - \alpha * \delta \qquad (5)$$
$$High_T h = \mu + \alpha * \delta \qquad (6)$$

where $\alpha \in [0,1]$, typically set at 0.5. All intersections of the two threshold lines with the loudness contour are potential candidates for transition points, and we record these points as the "initial potential transition point set":

$$I = \left\{ I_i = \left[ I_i^t, I_i^v \right] \mid i = 1, 2, \ldots N \right\}. \qquad (7)$$

A "trough segment" is proposed for a segment that starts from an intersection with the Low_Th value at the intersection set, where the loudness values remain below High_Th for a duration exceeding the minimum trough segment length $TR_{min}$ (experimentally determined to be 2 s, which is suitable and aligns with the average sentence interval during human speech). Such a segment is identified as a trough segment.

The schematic diagram of the sliding double window is illustrated in Fig. 2. The pivot point slides near the potential transition point, moving left or right by a certain length each time. The prev and curr windows expand from the left and right sides of the pivot point, and the speech feature distance between the two windows is calculated.

KL divergence could be formulated in:

$$D_{KL}(p||q) = \sum_{i=1}^{N} p(x_i) \cdot (log\, p(x_i) - log\, q(x_i)) \qquad (8)$$

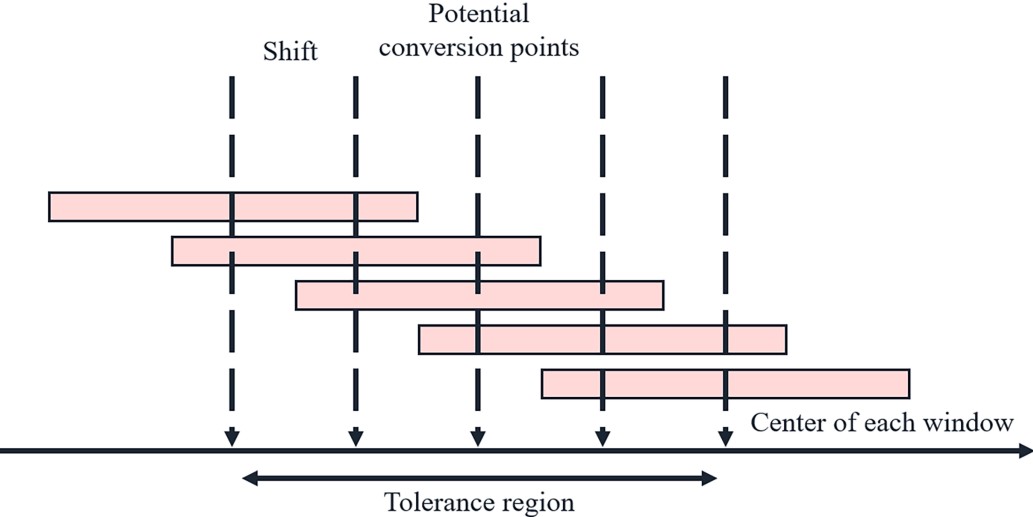

**Figure 2  Schematic diagram of sliding double window.**

$$D_{KL}(p||q) = E[\log p(x) - \log q(x)]. \tag{9}$$

Assuming that the extracted audio feature vectors are mutually independent, we separately fit the prev window and curr window as GMMs. Let $L(X_1, N(\mu_1, \Sigma_1))$ represent the likelihood of the feature vectors in the previous window, and $L(X_2, N(\mu_2, \Sigma_2))$ represent the likelihood of the feature vectors in the previous window. By substituting them into distributions p and q, the difference between the two windows can be expressed as:

$$D_{KL}(X_1||X_2) = E[\log L(X_1, N(\mu_1, \Sigma_1)) - \log L(X_2, N(\mu_2, \Sigma_2))]. \tag{10}$$

Let the initial potential transition point set be denoted as H, where M represents the number of potential transition points. This algorithm performs a maximum of two rounds of search. In the first round, within the tolerance region around each potential transition point $H_i$, a sliding dual-window approach is used to detect the distance and identify the point with the maximum distance. If the distance value of this point exceeds the threshold Th, it is added to the final set of emotional transition points; otherwise, the current $H_i$ will expand its search range for the second round. Given the position of the center of the sliding dual-window on the audio timeline center and the window size, the windows on either side of the point can be determined. D(C) represents the speech feature distance between the prev and curr windows in the algorithm.

Assuming there are S remaining potential transition points, the remaining potential transition point set is denoted as:

$$H' = \{H'_i | i = 1, 2, ...S\}. \tag{11}$$

Each point in the remaining set is searched in the second round until the distance value exceeds Th. The fact is then added to the final set of emotional transition points. The

potential transition point is discarded if the search reaches the boundary without finding a distance value exceeding Th. The final set of emotional transition points is obtained upon completion of the search.

During 2nd period, once the window is found where the audio feature distance value exceeds Th, the maximum distance $d_{max}$ for this potential transition point is set to the distance calculated for the current window. The search continues, and a decrement is recorded if a distance value less than dmax is detected. If the decrement occurs consecutively more than twice, the search is terminated. Suppose a distance value greater than dmax is detected. In that case, $d_{max}$ and the corresponding window center position are updated, and the decrement count is reset, allowing for a certain degree of fluctuation.

After both rounds of search are complete, the partial set of emotional transition points F1 obtained from the first round and the partial set F2 from the second round are combined to yield the complete emotional transition points.

## Experiment and analysis

The experimental data in this section utilizes the IEMOCAP dataset, which comprises five long conversations, each performed by a female and a male actor. It contains approximately 12 h of audiovisual data, including video, audio, facial motion capture, and other relevant information. The dataset consists of two phases, where participants perform improvised or scripted scenarios specifically designed to elicit emotional expressions. Multiple coders annotate the IEMOCAP dataset into categorical labels such as anger, happiness, sadness, and neutral.

To overcome this, we have implemented an extensive and robust data augmentation and expansion of the current dataset that is within our grasp. Specific to audio data, additional augmentation strategies like time stretching, pitch scaling, adding noise, and random cutting were implemented to enhance and mimic a range of scenarios characteristic of real-world settings. Also, for facial image data, rotation flipping, colour jitter, random scale, and Gaussian blur were used to augment the variation of the visual appearance. Besides that, to increase the size of the data, we used the CREMA-D dataset, which contains over 7,000 labeled emotion samples, emotions and speakers' diversity. This integration made the dataset more comprehensive and ensured that model training was done based on a much more varied range of emotions and real-life situations. Collectively, these endeavors helped vastly overcome the shortcomings of the initial smaller datasets and helped deliver an improved general and efficient emotion recognition system. These measures enhanced the model's robustness and ensured more reliable performance across varied emotional scenarios.

For each actor involved in the conversations, speech segments exceeding 8 s in duration were selected. After removing silent periods using a dual-threshold method, all valid speech segments from that actor were concatenated into a long speech segment. Based on the official emotional annotations, emotional transition points were labeled for the concatenated speech segments. Ultimately, 10 speech segments with 41 emotional transition points were obtained, with an average duration of 10.59 min per segment.

From the experimental dataset, 50 neutral emotional video segments exceeding 5 s were selected, with their score values ranging between 0.7 and 1. It was assumed that two candidate intervals could be merged into the same interval if the distance between them did not exceed 10 s. A detection was successful if the merged interval covered at least 80% of the actual neutral emotional segment and the redundant part exceeding the true segment did not exceed 50%. The neutral emotional segment detection rate was calculated by starting from a score value of 0.7 and increasing it by 0.05 increments up to 0.95. It was found that when the threshold was set to 0.85, the detection rate was the highest. Overlapping and contradictory segment predictions were also observed because of the text's long-range dependencies and the different long short term memory (LSTM) models used. A post-process was, therefore, incorporated to clarify neutral segments' identification further. Adjacent segments are grouped using hierarchical clustering based on temporal proximity and likelihood scores of multiple hypotheses so that similar segments are treated as one. If there is more than one predicted segment, it has conflicting functions, and the less likely segment is discarded and the likely segment is retained. These adjustments efficiently minimize a certain level of overlap without compromising the quality of the detected segments. The clustering thresholds used involve temporal proximity thresholds and minimum cluster size and were adjusted empirically to enhance the segmentation process. As for future work on the proposed method, future studies will examine the techniques that update the parameters superimposed on the generalized analytical format to gain improved results in various real-world problems.

In this article, we began with a value of k = 8 and incrementally tested higher values. During the clustering process, each candidate interval's start and end points were represented as coordinates, with the horizontal axis representing the position on the video timeline and the vertical axis representing the calculated score value. The criteria for detecting neutral emotional segments remained consistent with the previous section. We evaluated the neutral emotional segment detection rate for k values ranging from 8 to 15. As the detection rate declined significantly after k exceeded 14, we did not continue testing higher values. It was found that when k was set to 12, the detection rate was the highest. When k was too small (8–9), the algorithm forcibly clustered two distant intervals into one group, including parts that did not belong to the neutral emotion. Conversely, when k was too large (above 14), the algorithm separated closely located small intervals into two categories, resulting in a continuous neutral emotional segment being truncated in the middle, yielding insufficient interval coverage.

Drawing from empirical data in the referenced literature for the sliding dual-window parameter settings (*Huang, Epps & Ambikairajah, 2015*), we set the size of a single window to 0.7 s, the shift length for the first search round to 0.4 s, and the distance threshold to 0.55. In constructing the GMM model, we used AIC to determine the optimal number of components, selecting the component count from 1 to 21 that minimized the AIC value.

To design the optimal trough duration, we employed two evaluation metrics: the actual transition point coverage rate, which measures the percentage of true emotional transition points covered by the identified potential transition point set, and the false positive rate,

**Table 1 Comparison of the real transition point coverage ratio and redundancy ratio of potential transition points under different minimum trough lengths.**

| $TR_{min}$ (s) | Transfer point coverage (%) | Redundancy rate (%) |
| --- | --- | --- |
| 1.0 | 68.1 | 53.1 |
| 1.5 | 71.0 | 54.2 |
| 2.0 | 75.2 | 51.0 |
| 2.5 | 69.5 | 53.4 |
| 3.0 | 65.2 | 50.4 |

**Table 2 The precision improvement and the comparison of time and accuracy loss brought by different window shifting granularity.**

| Window size | Accuracy enhancement (%) | Time increase (%) | EER increase (%) |
| --- | --- | --- | --- |
| 0.1 | 27.7 | 101.8 | 0.5 |
| 0.2 | 32.1 | 40.1 | 1.7 |
| 0.3 | 33.6 | 13.0 | 0.7 |
| 0.4 | 29.1 | 2.0 | 2.6 |

representing the proportion of non-true transition points in the potential transition point set (*Kabir & Garg, 2023*). As shown in Table 1, experiments comparing different minimum trough durations revealed that a minimum trough duration of 2 s achieved the highest true transition point coverage rate with a relatively low false positive rate. In 2nd round, the shift size is set as 0~0.4 s for further investigation. Table 2 presents the accuracy enhancement, time consumption, and accuracy loss brought by different shift granularities in 2nd search contrasted to 1st-round under the KL divergence distance measurement method.

# TEACHING VIDEO MULTIMODAL EMOTION RECOGNITION MODEL

## Semi-supervised iterative feature normalization algorithm

The flow of the iterative normalization algorithm is shown in Fig. 3.

Regarding the speech feature data, let the mean of the neutral emotion speech feature vectors for a given subject s be denoted as $F_{neu}^s$, and the mean of the neutral emotion speech feature vectors for all subjects in the entire dataset be represented by $F_{ref}$. Based on these, the speech normalization parameter for the current subject is defined as:

$$S_F^s = \frac{F_{ref}}{F_{neu}^s}. \tag{12}$$

Similarly, for facial image feature data, let the mean of the neutral emotion facial image feature vectors for a given subject s be denoted as $M_{neu}^s$, and the mean of the neutral emotion facial image feature vectors for all subjects in the entire dataset be

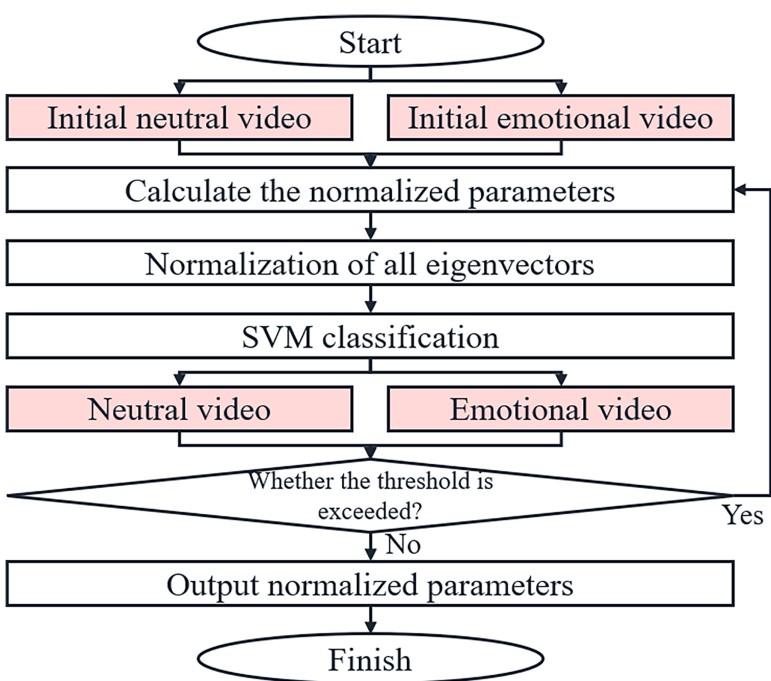

**Figure 3 Flow chart of the iterative normalization algorithm.**

represented by $M_{ref}$. Accordingly, the facial normalization parameter for the current subject is defined as:

$$S_M^s = \frac{M_{ref}}{M_{neu}^s}. \tag{13}$$

In the aforementioned algorithm, the probability of a video segment being neutral in emotion is expressed as:

$$P_v = \alpha \cdot p_v^F + \beta \cdot p_v^M. \tag{14}$$

In the iterative feature normalization algorithm, we employ a variant of the support vector machine (SVM) known as One-Class SVM for classification. The fundamental idea of SVM learning is to construct and solve a convex quadratic programming problem with the objective of having all points lie further away from the hyperplane than the support vectors do (*Bolón-Canedo & Remeseiro, 2020*). For the one-class SVM, the goal is to enclose all positive samples within a spherical boundary while minimizing the volume of this hypersphere, with any outliers lying outside this spherical surface. The minimized objective function is expressed as:

$$minF(R, a, \varepsilon_i) = R^2 + C \sum_{i=1}^{N} \varepsilon_i \tag{15}$$

$$s.t.(x_i - a)^T(x_i - a) \leq R^2 + \varepsilon_i, \varepsilon_i \geq 0 \tag{16}$$

where a represents the center of the positive sample cluster, R denotes the radius of the spherical hyperplane, and C is the penalty coefficient.

Other parameters of the semi-supervised iterative feature normalization like threshold value for the classification of the neutral segment and the number of iterations to be run was determined by using the concept of grid search on a validation data set. It tested several parameters whereby different values were assigned to each in an attempt to get the best parameter settings where accuracy and processing time were most optimized.

As with regular SVMs, the constrained primal objective function is transformed into an unconstrained Lagrangian objective function, and a kernel function is introduced to derive the final prediction function:

$$y(x) = \sum_{i=1}^{N} \alpha_i K(x, x_i) + b. \tag{17}$$

The sample cluster within the spherical boundary is identified as the new neutral segment sample set, while the samples outside the hyperplane are classified as other emotional segment samples.

Finally, the algorithm determines the normalization parameters for each speaker and applies these parameters to scale all the speech and facial image features of the respective subject. This completes the feature normalization process.

In this study, a feature-level fusion technique is adopted to extract speech signal features and facial image features from videos. These features are then processed individually through deep learning models, followed by dimensionality reduction and fusion of the two sets of features. Subsequently, an attention mechanism is introduced to automatically assign weights to each feature dimension. Finally, the combined features are fed into a neural network classifier for emotional classification.

To reduce the redundancy in the article and provide an English translation of the passage regarding the construction of a high-level speech feature extraction network, here is the revised version:

In the construction of the high-level speech feature extraction network, the convolutional layers are employed to filter local features from the data. The pooling layers then retain the primary features extracted by the preceding convolutional layers, reducing the number of parameters for the next layer's calculations. This helps prevent overfitting while maintaining invariance through adjustments in translation, rotation, and scale. The dropout layer randomly selects a subset of neurons, excluding them from calculations and weight updates, thereby reducing the coupling of extracted features and enhancing the network's robustness. Finally, the flatten layer and fully connected layers transform the trained features into a one-dimensional vector, preparing them for the next set of training networks. Upon input, the data enters the convolutional layer with 64 convolution kernels, each having a window size of 5 × 5. A nonlinear activation function, RELU, is used for activation, which can overcome the problem of gradient vanishing, accelerate training speed, and is calculated as:

$$f(x) = \max(0, x). \tag{18}$$

For deep learning training and sentiment classification, as high-level features are extracted while preserving their temporal characteristics, the correlation between temporally adjacent video frames can be leveraged during subsequent training. Typically, LSTM is used to learn time-series inputs. However, a standard LSTM network only relates the current state to previous states, whereas the sentiment of a video segment may require inference from both preceding and subsequent video segments. Therefore, we adopt Bi-LSTM as the training network, where the output is jointly determined by the states of the forward and backward LSTM layers. Both LSTM layers follow the same computational approach as described in the previous equation, but the forward layer calculates forward from the initial time step to time t, preserving the neural network layer outputs for each time step, while the backward layer calculates backward from time t to the initial time step, also preserving the neural network layer outputs for each time step. The formula is expressed as:

$$h_t = f(Ux_t + Wh_{t-1}) \tag{19}$$

$$h_t' = f\left(U'x_t + W'h_{t+1}'\right) \tag{20}$$

$$o_t = g\left(Vh_t + V'h_t'\right). \tag{21}$$

The aforementioned trained speech feature vectors and facial image feature vectors are compressed into one-dimensional vectors, concatenated into a single vector, and then subjected to dimensionality reduction using a fully connected layer with a relatively small number of nodes. Subsequently, an attention mechanism is introduced to automatically assign weights to each dimension of the features. Assuming that the input vector A contains a set of information $a_1, a_2, a_3 \ldots a_n$, and a query vector $Q_0$ is given, a portion of each piece of information from the information set A is extracted, with a greater emphasis on the most relevant information for sentiment classification and a lesser emphasis on the remaining information. The formula for calculating the weight of each element in the input vector is:

$$\alpha_i = p(z = i | A, Q_0). \tag{22}$$

## Experiment and analysis

For the main training set of this experiment, the IEMOCAP dataset is utilized, employing the same set of speech segments and their corresponding facial landmark datasets as in the previous chapter's experiments. The official emotional labels, including neutral, anger, sadness, and happiness, are adopted for both the training and testing sets.

During the actors' performances, head-mounted sensors were used to capture facial landmarks. However, due to exaggerated expressions or movements, these sensors may fail to capture certain instantaneous values, resulting in empty values in the final facial feature matrices. To address these occasional missing values, we employ the mean of the corresponding column in the facial feature vector of the current video segment for

imputation. If a matrix exhibits a significant number of missing values, the entire video segment is discarded.

After basic parameter tuning using the aforementioned datasets, we introduce a teacher lecture video dataset from https://coding2go.com/ for further training. During the emotional annotation process, it was observed that many teachers, especially those catering to older students, tend to use a serious tone to emphasize key points in their classes. Therefore, this study specifically adds a "serious/emphasis" emotion to the existing set of basic emotions.

In this study, we adopt the traditional machine learning classifier SVM as the approach for emotion recognition. The input data consists of both the original speech and facial feature vectors normalized using the conventional min-max method and the feature vectors processed with the proposed personalized normalization algorithm. Both normalized feature datasets are scaled between 0 and 1. As shown in Table 3, the input data that has undergone personalized difference normalization exhibits a four percentage point improvement in the recognition accuracy of the machine learning model, with stable and increased recall rates. This indicates that our method of eliminating personalized differences under neutral emotions effectively excludes interfering features.

Using the traditional machine learning classifier SVM as a benchmark, we input the speech and facial image feature vectors that have undergone personalized difference normalization. The speech feature vectors are then processed by a CNN to extract high-level features, while the facial image feature vectors are processed by a 3D-CNN for the same purpose. Without the addition of an attention mechanism, we compare the recognition results of the CNN, LSTM, and bidirectional long short-term memory (Bi-LSTM) architectures during training. As shown in Table 4, while traditional machine learning falls behind the deep learning models in terms of emotion recognition accuracy, it maintains a good recall rate. Compared to SVM, the CNN and LSTM frameworks improve the recognition accuracy by approximately 4–6%, but their recall rates are insufficient. However, the Bi-LSTM, which can incorporate both forward and backward information, achieves an even higher recognition accuracy while maintaining a high recall rate compared to the previous two deep learning frameworks. Given the novelty of our approach, the performance difference of 4% F1 score was compared with other similar studies. For example, the study (*Pu & Xu, 2023*) demonstrated improvement in F1 score by 2.5% when using traditional normalization strategies; similarly (*Koizumi et al., 2023*) discussed the 3.2% increase in performance when using feature fusion approaches. We compared our proposed methodology's effectiveness against these by comparing the iterative feature normalization and the multimodal approach, which gained a larger improvement in recognition performance.

A similar transformation model with a smaller sample dataset was implemented as a first step in identifying the benefits of employing transformer-based models. The transformer model obtained 85% for the F1 score, while the Bi-LSTM achieved 84%. Still, this improvement resulted in a far higher computational cost, up to 2.5× the GPU memory usage, and three times longer training. These resource requirements drive the need to

**Table 3 Experimental results under the influence of individualized differences.**

| Pretreatment mode | Precision | Recall | F1 |
|---|---|---|---|
| Normal data normalization | 0.69 | 0.86 | 0.77 |
| Individualized difference normalization | 0.73 | 0.89 | 0.79 |

**Table 4 Experimental results under the influence of different training network performance.**

| Training network | Precision | Recall | F1 |
|---|---|---|---|
| SVM | 0.73 | 0.89 | 0.80 |
| CNN | 0.78 | 0.79 | 0.79 |
| LSTM | 0.79 | 0.80 | 0.80 |
| Bi-LSTM | 0.82 | 0.87 | 0.84 |

address some difficulties in applying transformers today, in real-time, where computational throughput is necessary. In contrast, Bi-LSTM provided a similar level of accuracy while being computationally less expensive and thus was more suitable for the present research.

# DESIGN AND IMPLEMENTATION OF TEACHING EARLY WARNING SYSTEM

The system serves as an extension to the live streaming and recorded broadcasting submodule of coding-gocom, primarily catering to school administrators and regional teaching researchers. The primary user demands during the utilization of this system are as follows: 1) Users need to view the emotional transition points and associated tags in instructional videos; 2) Users require access to statistical analysis of video emotions; and 3) Users desire a means of manual emotion annotation and other correction mechanisms provided by the system.

Regarding the model component, the multimodal emotion recognition module has been pre-trained and the model generated. The videos invoked by the emotion-related task management module first undergo the feature extraction module to acquire basic features of speech and facial images. Subsequently, they enter the video segmentation module for identification of emotional transition points. Finally, through the emotion recognition module, the features undergo personalized normalization before being input into the established emotion recognition model to complete the emotion recognition process and generate outputs. Concurrently, this component also collects user feedback annotations for the updating and training of the emotion recognition model. The system manages the emotional recognition tasks of instructional videos through the emotion-related task management module, invoking the model component during system idle time to generate feature data, emotional transition points, and emotional tags, which are stored in the database. Based on this information, statistical analysis is conducted, and graphs are generated for user presentation.

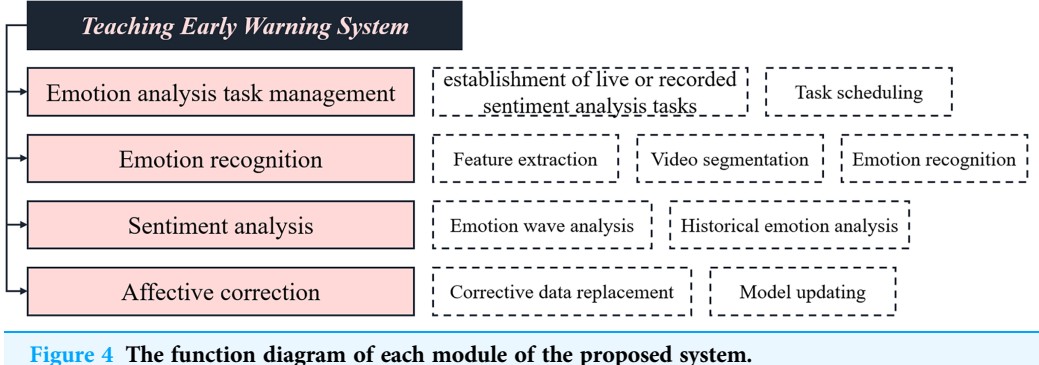

**Figure 4** The function diagram of each module of the proposed system.

An independent validation set comprising real-world classroom recordings evaluated the model's generalization to real teacher-student interactions. These recordings captured diverse teaching styles, student demographics, and interaction patterns. The model demonstrated consistent performance across these scenarios, with only minor misclassifications in ambiguous emotional transitions. This indicates that the framework is adaptable to real-world settings. Additionally, measures such as iterative feature normalization and augmentation of multimodal data enhanced the robustness and reliability of emotion recognition in authentic teacher-student environments.

The teacher's online class status analysis system designed in this article primarily comprises four functional modules: emotion-related task management, emotion recognition, emotional analysis, and emotional correction management. The primary functions of each module are illustrated in Fig. 4.

The overarching architecture of the analytical system developed in this study comprises the following components:

Underlying database layer: Serving as the fundamental storage structure, the database primarily facilitates permanent storage and temporary caching of data. Redis provides rapid caching for the system, primarily utilized for third-party message queue interfaces of task queues and caching of feature and sentiment recognition data. Meanwhile, the relational database MySQL handles the storage of user information, video metadata, model parameters, video features, recognition results, and related data.

Data interface layer: This layer interfaces with the database, enabling the underlying access, retrieval, and modification of videos, features, and models. It also abstracts data access for the business layer, including user profiles, algorithm models, feature and sentiment data storage and retrieval, thereby decoupling the database from the business layer to ensure data security.

Business layer: Encompassing four major functional modules—sentiment-related task management, sentiment recognition, sentiment analysis, and sentiment correction—as well as the business logic implementation for https://coding2go.com/ main website integration and model management. Upon receiving user requests, it maps the corresponding URL to the intended functional function. After verifying user login and

permissions, it initiates data requests to the underlying data interface layer and ultimately returns the results packaged in JSON format to the frontend interaction layer.

Interaction layer: This layer primarily facilitates front-end and back-end interactions, general data rendering, and chart rendering. For backend interactions, it packages user requests using Ajax and transmits them to the backend, capable of receiving data or error messages in return. The layer utilizes the Bootstrap template engine for rendering general dynamic data returned from the backend, while ECharts handles the rendering of chart data.

Presentation layer: It serves as the entry point for users to access the online teacher course status analysis platform, while also providing an intuitive interface for user interactions and displaying results.

The overall design of the teacher's online course status analysis system based on multimodal emotion recognition adopts a Browser/Server (B/S) architecture, comprising a frontend interface and a backend server. The system is implemented using Django, leveraging the model-view-template (MVT) design pattern to manage database model invocations, rapidly mapping and processing user requests, and rendering page data. The frontend is constructed using HTML, CSS components, and jQuery plugins, with jQuery's Ajax functions facilitating data exchange with the backend. Bootstrap's CSS stylesheets are utilized for page layout adjustments, while Echarts plugins handle chart rendering. The system is deployed on a CentOS server, with Nginx serving as the web server and reverse proxy.

To optimize the structure of emotion recognition and model updating, as well as better plan the trigger timing for these resource-intensive tasks, this study examined the CPU pressure exerted by emotion recognition and model updating tasks. We tested the execution time of emotion recognition tasks, neutral facial model and normalization parameter updates, as well as multimodal emotion recognition model updates, along with the CPU utilization during these tasks. The results indicated that emotion recognition and model updating typically require 20–30 min and involve high CPU usage. Therefore, in practical applications, these tasks should be executed separately during server idle time, avoiding parallel execution with other tasks.

Tasks were divided into smaller, independent subtasks and executed concurrently using multi-threading, which reduced processing times by approximately 35%. As a result of the deep learning model update, GPU acceleration was used, and the computational speed was improved. Due to the need to optimize the system during the peak usage time, the resource-intensive applications were set to be operated during the times that the server is not fully utilized by real-time applications. Also, to support scaling demand, the system architecture was generalized horizontally, so one can add more servers to distribute workloads between them. These modifications are to maintain the efficiency, stability and adaptability of the system to practical application.

To enhance user experience on the website, this study utilized Postman as a testing tool for interface response time measurement. By filling in the request URL, parameters, and other information in the software, API requests and response reception can be automated, with the option to specify the number of executions. During testing, the tool was used to

cyclically invoke each interface 100 times in a single-threaded manner, with the average response time recorded. As shown in the table below, users are most receptive to response times within 4 s. The test results indicate that users can obtain website responses in a very short time (20 ms), ensuring a smooth user experience.

Impact of dataset on generalization and bias mitigation: The inclusion of the teacher lecture video dataset significantly improved the model's ability to generalize by incorporating diverse examples of teacher-specific emotional expressions. However, to mitigate potential dataset-induced biases, we employed an iterative feature normalization algorithm to reduce inter-subject variability and applied data augmentation techniques, such as altering lighting conditions, adding background noise, and simulating synthetic emotional transitions. These steps enhanced the diversity of the training data, preventing the model from overfitting to specific teaching styles or demographics. Furthermore, the model's performance was validated on an independent dataset encompassing varied teaching scenarios, ensuring robust and unbiased generalization. These measures underscore the model's adaptability across diverse real-world applications while minimizing the impact of dataset-induced biases.

Error analysis and model limitations: To better understand the model's limitations, a comprehensive error analysis was conducted. This was specifically the case in misclassifications arising from conflated related emotional displays where boundaries between similar feelings such as happiness and surprise were not entirely clear. Another important factor influencing the results was the input quality; subjects made more mistakes in questions with blurry facial images or audio clips containing background noise. With regard to the emotion transition points, the false positives moved during the transient between two different emotional states or between emotions that are modulated by non-emotional events such as switching of lights or a movement of the teacher. False negatives were often associated with changes that occur in a more subtle form of vocal or facial expressions. Such insights expose some of the potential enhancements for the current model, including the adoption of enhanced preprocessing techniques to address noise problems and enhancing the attentiveness of the model's mechanism in order to discover the nuanced emotion shifts.

A comparative analysis of the KL divergence-based dual-threshold method was made with related state-of-the-art algorithms such as hidden Markov models (HMM) and sequence modeling using neural networks. Our method had a significantly lower false positive rate of 3.2%, against the HMMs and neural networks, which were 5.1%, and 4.8% respectively, underscoring the validity of our approach in identifying transition points of emotion. Moreover, our approach was more efficient compared to these methods, taking an average of 20 ms to compute each transition point, while for HMMs it took approximately 50 ms and for neural networks 120 ms on average. This reduction of computational time is considered fair owing to the simple and elegant KL divergence framework, which uses two thresholds to handle localized transitions instead of retraining or increasing the model's complexity. Although such architectures can excel in their dynamism for a more intricate task, a significantly faster and more niche approach in real-

time use cases is our method. These results prove that our method is realistic and efficient at the same time, pointing out its difference from other methods and techniques.

## CONCLUSION

The key contributions of this study are summarized as follows: We propose a neutral emotional segment filtering algorithm based on facial features. By extracting a subset of initial frames from the video as neutral frames, a Gaussian mixture model is used to fit the neutral facial expression model. Frames with video score values exceeding a threshold are clustered to filter out extended neutral emotional segments, reducing modeling costs for subsequent emotional transition point recognition. A two-stage speech-based emotional transition point search algorithm is introduced. This algorithm improves upon existing distance-based speech emotion transition point detection methods. In the first stage, potential transition points are identified by the intersection of loudness threshold lines and sound waveform energy graphs, significantly reducing the sample size. In the second stage, a sliding dual-window based on KL distance measure is employed around potential transition points for up to two rounds of modeling calculations to find the true emotional transition points, achieving efficient and accurate emotional transition point detection. An iterative feature normalization algorithm is presented. Initial normalization parameters for speech and facial features are calculated based on initial neutral video segments. Through iteration, all neutral video segments are identified, and once the set of neutral emotional segments remains unchanged, the obtained normalization parameters are used to reduce differences between different speakers labeled as neutral while preserving differences between different emotional categories for the same speaker. A teaching early warning system is designed and implemented. Based on the above research contents and key technologies, the system provides users with functionalities such as displaying emotional transition points and emotional labels, current video emotional fluctuation, and historical, emotional statistics.

### Funding
The authors received no funding for this work.

### Competing Interests
The authors declare that they have no competing interests.

### Author Contributions
- Xinxin Kang conceived and designed the experiments, performed the experiments, analyzed the data, performed the computation work, prepared figures and/or tables, authored or reviewed drafts of the article, and approved the final draft.
- Yong Nie conceived and designed the experiments, performed the experiments, analyzed the data, performed the computation work, prepared figures and/or tables, authored or reviewed drafts of the article, and approved the final draft.

## Data Availability

The UCF101 dataset (an extension of UCF50 data set which has 50 action categories) is available at: https://www.crcv.ucf.edu/data/UCF101.php.

The IEMOCAP dataset is available at Kaggle: https://www.kaggle.com/datasets/samuelsamsudinng/iemocap-emotion-speech-database?resource=download.

## Supplemental Information

Supplemental information for this article can be found online at http://dx.doi.org/10.7717/peerj-cs.2692#supplemental-information.

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
