# Peer review of "Design and analysis of teaching early warning system based on multimodal data in an intelligent learning environment"

_PeerJ Computer Science, doi:10.7717/peerj-cs.2692_

## Round 0.1 · original submission · Major Revisions

Dear Authors,

The reviews for your manuscript are included at the bottom of this letter. We ask that you make changes to your manuscript based on those comments. Reviewer 1 has asked you to provide specific references. You are welcome to add them if you think they are useful and relevant. However, you are not obliged to include these citations, and if you do not, it will not affect my decision.

Best wishes,

Reviewer 1 ·

Basic reporting

- Some terms, like “neutral emotional segment,” are defined too late in the text, potentially confusing readers. A glossary or earlier clarification would improve comprehension.

- The results section mentions performance improvements (e.g., a 4% F1 score increase) but doesn't benchmark these gains against similar studies. Adding such comparisons would highlight the novelty of the findings.

Experimental design

- The use of IEMOCAP and teacher video datasets is appropriate but limited. The dataset size (~12 hours) is insufficient for training deep learning models robustly. Augmentation or expansion of datasets should be explored.

- Methods for selecting optimal parameters (e.g., Gaussian components for GMMs, window size for dual-threshold methods) are not entirely transparent. Additional details would make replication more feasible.

Validity of the findings

- The paper does not thoroughly analyze misclassifications or sources of error (e.g., false positives in emotion transition points). A detailed error breakdown would provide insights into model limitations and areas for improvement.

Additional comments

This study proposes a multimodal emotional recognition model and teaching early warning system to analyze teacher emotions in online instructional videos, using algorithms for emotional segment filtering, transition point detection, and iterative feature normalization. The system, incorporating deep learning and attention mechanisms, enhances online education by providing actionable insights into teacher-student interactions and emotional dynamics. However, the paper suffers from the limitations listed below, which must be “fully” addressed before its reconsideration:

1- The study claims high accuracy in detecting emotional transition points using KL divergence with dual thresholds. How does this compare with alternative methods like HMM or neural network-based sequence modeling in terms of false positive rates and computational time?

2- Given that the IEMOCAP dataset includes only 12 hours of audiovisual data, how can the findings generalize to more extensive datasets? Would additional datasets improve the robustness of the model, and why was the chosen dataset considered sufficient?

3- The iterative feature normalization algorithm relies on one-class SVMs for neutral emotion classification. How does this approach compare in terms of scalability and performance with newer techniques like autoencoders for anomaly detection?

4- The model updates and emotion recognition tasks reportedly take 20-30 minutes with high CPU usage. Were any efforts made to optimize or parallelize these tasks, and how scalable is the system for real-world applications?

5- The study reports that Bi-LSTM achieves the highest recognition accuracy (84% F1 score). How does this compare with transformer-based models, which are increasingly common in sequence modeling?

6- The authors optimized their deep learning model using feature normalization, attention mechanisms, Bi-LSTM networks, dimensionality reduction, diverse datasets, imputation of missing data, and hyperparameter tuning. This is valuable, however, apart from these techniques, reducing dataset Kolmogorov complexity (the shortest program to produce the output) can also improve model accuracy, as seen in studies leveraging decreased complexity for rapid accuracy gains (e.g., https://doi.org/10.1007/s10462-019-09750-3; https://doi.org/10.1038/s41598-023-28763-1). Please read and reference these two papers.

7- The detection rate for neutral segments is highest at a threshold of 0.85, but redundancy is not fully addressed. How does the method deal with overlapping or contradictory segment predictions in real-world scenarios?

8- The authors supplemented training with data from a teacher lecture video dataset. How did this influence the model's generalization ability, and were there any measures to mitigate dataset-induced biases in emotion recognition?

Reviewer 2 ·

Basic reporting

This manuscript is fairly written. However, the following suggestions can strengthen the manuscript.

1- The language should be more clear, professional, and accessible to the target audience.

2- Authors should provide more details related to Tables and figures.

Experimental design

1- The authors used semi-supervised iterative feature normalization, there is a need to specify how hyperparameters were selected or optimized.

2- Authors should provide information about how well the model generalizes the real teacher-student interactions.

3- please justify the selection of Bi-LSTM model, where as more advanced models exist such as Transformer.

4- How the robustness of the model was assessed?

Validity of the findings

1- Please provide the information on how the threshold was validated.

---

## Round 0.2 · accepted · Accept

Dear Authors,

It is to be commended that you have addressed the comments made by the reviewers and improved the quality of your work. Your manuscript seem ready for publication.

Best wishes,

Reviewer 1 ·

Basic reporting

The authors addressed my comments.

Experimental design

The authors addressed my comments.

Validity of the findings

The authors addressed my comments.

Additional comments

The authors addressed my comments; thus, the paper can be accapted in the present format.

Reviewer 2 ·

Basic reporting

This paper can be accepted. The depth of analysis and the innovative approach reflected in the work are commendable.

Experimental design

I am impressed by the novelty of the research and its potential to inspire further exploration in the field.

Validity of the findings

The clarity in the presentation and the commitment to advancing knowledge in this area are highly admirable.